# Model-based and phylogenetically adjusted quantification of metabolic interaction between microbial species

**Tony J. Lam, Moses Stamboulian, Wontack Han, Yuzhen Ye** *

Luddy School of Informatics, Computing and Engineering Indiana University, Bloomington, IN, USA

* yye@indiana.edu

**Data Availability Statement:** Implementation and data are available at https://github.com/mgtools/PhyloMint.

**Funding:** This research was supported by the National Institute of Health [1R01AI108888 to Y.Y.,

## Abstract

Microbial community members exhibit various forms of interactions. Taking advantage of the increasing availability of microbiome data, many computational approaches have been developed to infer bacterial interactions from the co-occurrence of microbes across diverse microbial communities. Additionally, the introduction of genome-scale metabolic models have also enabled the inference of cooperative and competitive metabolic interactions between bacterial species. By nature, phylogenetically similar microbial species are more likely to share common functional profiles or biological pathways due to their genomic similarity. Without properly factoring out the phylogenetic relationship, any estimation of the competition and cooperation between species based on functional/pathway profiles may bias downstream applications. To address these challenges, we developed a novel approach for estimating the competition and complementarity indices for a pair of microbial species, adjusted by their phylogenetic distance. An automated pipeline, PhyloMint, was implemented to construct competition and complementarity indices from genome scale metabolic models derived from microbial genomes. Application of our pipeline to 2,815 human-gut associated bacteria showed high correlation between phylogenetic distance and metabolic competition/cooperation indices among bacteria. Using a discretization approach, we were able to detect pairs of bacterial species with cooperation scores significantly higher than the average pairs of bacterial species with similar phylogenetic distances. A network community analysis of high metabolic cooperation but low competition reveals distinct modules of bacterial interactions. Our results suggest that niche differentiation plays a dominant role in microbial interactions, while habitat filtering also plays a role among certain clades of bacterial species.

## Author summary

Microbial communities, also known as microbiomes, are formed through the interactions of various microbial species. Utilizing genomic sequencing, it is possible to infer the compositional make-up of communities as well as predict their metabolic interactions. However, because some species are more similarly related to each other, while others are more

1R01AI143254 to Y.Y.] and the National Science Foundation [2025451 to Y.Y]. The funders had no role in study design, data collection and analysis, decision to publish, or preparation of the manuscript.

**Competing interests:** The authors have declared that no competing interests exist.

distantly related, one cannot directly compare metabolic relationships without first accounting for their phylogenetic relatedness. Here we developed a computational pipeline which predicts complimentary and competitive metabolic relationships between bacterial species, while normalizing for their phylogenetic relatedness. Our results show that phylogenetic distances are correlated with metabolic interactions, and factoring out such relationships can help better understand microbial interactions which drive community formation.

This is a *PLOS Computational Biology* Methods paper.

## Introduction

Recent advances in microbiome research have accelerated the study of the composition and function of microbial communities associated with different environments and hosts. Studies have shown the association of microbial communities with human health and diseases including type 2 diabetes [1], and efficacy of treatment including immunotherapy to cancers [2]. To reveal the mechanisms behind the microbiome-host interactions, it is important to understand how microbial species form communities and how the microbial communities interact with the host to mediate various biological processes [3].

Studying the principles underlying the structure and composition of microbial communities is of long-standing interest to microbial ecologists. The dynamics which govern microbial community assembly have been extensively debated, and it is disputed upon as to what extent the role of neutral or deterministic dynamics plays in microbial interactions [4, 5]. Some studies support the neutral hypothesis, which assumes that community structure is determined by random processes [6]. Other theories suggest that community assembly dynamics are govern by deterministic processes such as habitat filtering and niche differentiation [7, 8]. While many studies focus on species abundances to study community assembly, Bruke et al. [9] showed that the key level to address the community assembly may not lie at the species level, but rather the functional level of genes. While the aforementioned theories of community assembly may not be all-encompassing, they highlight varied dynamics which can contribute to community structure and affect the assembly of complex microbial communities.

Some studies have also shown that microbial communities tend to be more phylogenetically clustered than expected by chance, harboring groups of closely related taxa that exhibit micro-scale differences in genomic diversity [10–12]. In one such study, marine bacterial communities were observed at various locations and it was reported that local communities were phylogenetically different from each other and tend to be phylogenetically clustered [12]. However, some microbial communities have also shown the opposite patterns, in which taxa are less clustered and are less related than expected by chance [13, 14]. Together, these studies have explored the relationship between functional distances/metabolic overlap with phylogenetic relatedness, and they have given rise to competing theories of 'habitat-filtering' and 'niche differentiation': habitat filtering suggests that dominant species exhibit similar functional traits, whereas niche differentiation says that phylogenetically similar species are unable to co-exist due to similar traits and resource overlap [3]. Nevertheless, several methods have been developed for inference of bacterial interaction network based on the assumption that phylogenetically related species tend to co-exists. For example, Lo et al. [15] developed phylogenetic graphical lasso approach for bacterial community detection, based on the assumption

that phylogenetically correlated microbial species are more likely to interact to each other. Additionally, systematists have long argued that the comparison between species is not an independent process [16]; this is largely driven by the fact that related organisms share many genes and traits. The confounding effect of shared phylogeny has since inspired the development of methods and techniques, such as phylogenetically independent contrasts and phylogenetic generalized lease squares, to account for the dependent effect of phylogeny when comparing across species [17–19].

The study of microbial interactions and the dynamics which govern such interactions are important in providing insights to community assembly and ultimately processes which influence host health and disease. Insights into community complementarity and competition may also uncover symbiotic and antagonistic relationships and can be used to provide prospective candidates for probiotics. Leveraging the increasing availability of microbiome datasets, novel statistical and computational methods have been developed to infer bacterial interaction networks from co-occurrence information. Some examples include, SparCC [20], a tool to infer correlations by correcting for compositional data. Conner et al. demonstrated the importance of using null model to infer microbial co-occurrence networks [21]. Mandakovic and colleagues compared microbial co-occurrence networks representing bacterial soil communities from different environments to determine the impact of a shift in environmental variables on the community's taxonomic composition and their relationships [22]. MDiNE is another recently developed model for estimating differential co-occurrence networks in microbiome studies [23]. Notably, Faust et al. [24] applied generalized boosted linear models to infer thousands of significant co-occurrence and co-exclusion relationships between 197 clades occurring throughout the human microbiomes; their study revealed reverse correlation between functional similarity and phylogenetic distance among bacterial species, which is unsurprising. Despite of the numerous advances, it has been considered difficult to infer microbial community structure based on co-occurrence network approaches [25].

Functional profiles or biological pathways inferred from genomic sequences of the microbial species can provide mechanistic information about the functional traits of the microbes and potential cross-feeding. Genome-scale metabolic models (GEMS) can potentially provide mechanistic explanations to the association of bacterial species that are discovered by analyzing their co-occurrence in diverse microbial communities [26]. Many automated tools [27–30] have been made available for genome scale metabolic reconstructions (GENREs), however to get quality models these automated methods often require additional manual refinement including checks for stoichiometric consistency, defined media, and gap filling [31]. The challenges of manual curation often make it difficult to construct GEMs for a large consortium of microbes. Notably, Machado et al. [32] developed an automated tool called CarveMe, which uses a top-down approach to build species and community level metabolic models which the authors claim is able to produce comparable results to other tools while also reducing manual intervention [32, 33]. The ability to predict metabolic network of microbial members through GENREs has led some studies to focus on inferring levels and types of interaction among microbial species via metabolic models. Levy and Borenstein [34] introduced pairwise indices of metabolic interaction: the metabolic competition index and complementarity index, which are computed based on the overlapping and complementarity of the compounds that are contained in the metabolic models, respectively. By analyzing the metabolic interactions among 154 human-associated bacterial species and comparing the computed indices with observed species co-occurrence in microbiomes, the authors concluded that species tend to co-occur across individuals more frequently with species with which they strongly compete, suggesting that microbial assembly is dominated by habitat filtering [34]. Similar metrics have been introduced to quantify the

metabolic complementarity and competition between bacterial species, such as MIP (metabolic interaction potential) and MRO (metabolic resource overlap) [26].

By nature, two phylogenetically-close microbial species share similar functional profiles or biological pathways due to their genomic similarity. Additionally, co-evolutionary studies have also shown that comparative analyses between species cannot be assumed to be statistically independent, as comparative data of similarly related species correlate with each other due to shared ancestry [16, 35–37]. Thus, without factoring out the phylogenetic relationship (the confounding factor), any estimation of the competition and complementarity based on function/pathway profiles may be biased and cause problems in downstream applications. In this study, we focused on the large collection of human gut-associated genomes (including reference genomes and genomes assembled from metagenomic sequences, MAGs). We implemented an automated pipeline, PhyloMInt, for genome scale pathway reconstruction and for computing competition and complementarity scores based on the reconstructed pathways. Our results showed correlation between phylogenetic distance and metabolic competition/complementarity indices, indicating the importance of normalizing these indices by the phylogenetic distance between underlying microbial species. Using a discretization approach, we were able to detect pairs of bacterial species with complementarity scores significantly higher than the average pairs of bacterial species with similar phylogenetic distances. We further built a network of human-gut microbes based on complementarity and competition indices, and we discuss some of the results we derived by analyzing the network.

## Results

### Evaluation of the performance of GENREs on incomplete genomes

To assess the stability of CarveMe genome-scale metabolic reconstructions (GENREs) on incomplete MAGs, we simulated incomplete genomes by randomly removing clusters of neighboring genes from complete genomes and evaluated their resulting GENREs (See Methods). By comparing GENREs constructed from incomplete genomes to that of complete genomes, we observed that the distribution of the number of source and sink nodes remain relatively stable in respect to the number of removed genes (Fig 1, top two rows of panels). We further compared the similarity of the networks as measured by the overlap of the nodes and edges (Jaccard similarity) (Fig 1, bottom two rows of panels). We found that for most cases, the reconstructed metabolic networks of simulated genomes remain largely similar to those of the complete genomes, with actual differences smaller than the expected values of the differences that are proportional to the loss of CDSs (the red dotted lines in Fig 1). For example, the metabolic networks of simulated incomplete genomes of *E. coli* str K-12 MG1655 with only 80% of the total CDSs shared similar nodes and edges with the complete genome with Jaccard similarity greater than 0.9. Considering both the stability of metabolic networks generated from GENREs from incomplete genomes (as measured by the number of source and sink nodes, and the similarity of metabolic networks), and the fact that we only utilized near complete MAGs in our analysis, we believe that use of near-complete MAGS (>80% completeness) should have minimal impact on the calculation of the metabolic complementarity/competition indices. Results of simulations can be found in S1 Table.

### Impact of phylogenetic relationship on microbial complementarity and competition indices

We applied our pipeline to analyze 2,815 human gut related MAGs and computed their pairwise competition and complementarity scores (about 8M directed pairs). As shown in Fig 2A,

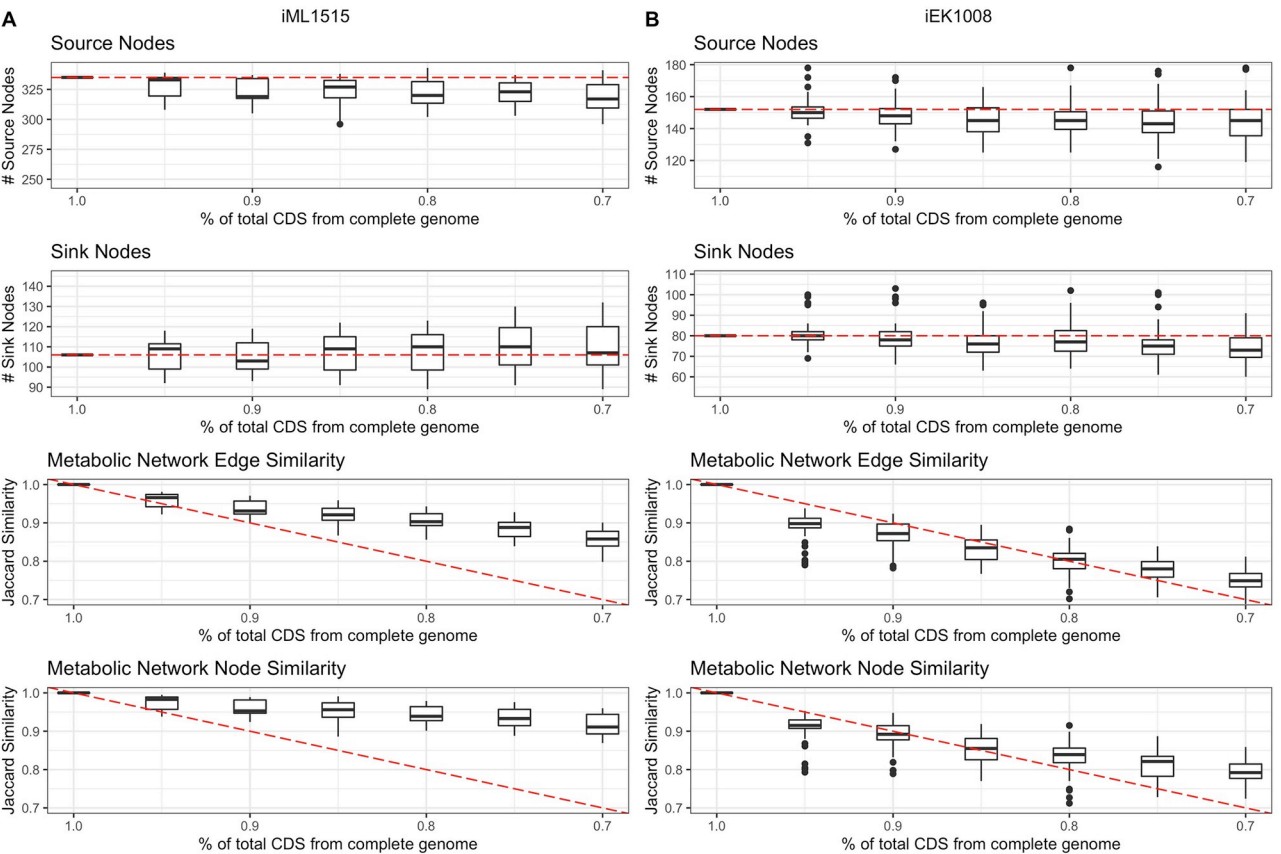

**Fig 1. Boxplots of the evaluation of CarveMe GENREs from simulated incomplete genomes.** Simulated incomplete genomes with 70%-100% of the total CDSs of the complete reference genome were used. Inferred metabolic networks of incomplete genomes were evaluated by the number of source nodes, the number of sink nodes, the overlap between the nodes of the networks (Jaccard similarity), and the overlap between the edges of the networks (Jaccard similarity). Dashed line in source and sink node boxplots represent the baseline number of source and sink nodes in a complete genome. Dashed line in metabolic network edge and node overlap boxplots represents a regression line with y-intercept of 1.0 and slope of -1; this line represents the expected value of Jaccard Index which is proportional to the total remaining CDSs. (A) GENRE of iML1515, *Escherichia coli str. K-12 substr. MG1655* (B) GENRE of iEK1008, *Mycobacterium tuberculosis H37Rv*.

we see a positive relationship between the metabolic complementarity of bacterial species and their phylogenetic distances. In contrast, we see in Fig 2B there is a negative relationship between metabolic competition of bacterial species and phylogenetic distance. Our results are consistent with other previous studies of functional and metabolic relationships with phylogenetic distances [24, 26, 38]. And they support the theory of niche differentiation, which states that phylogenetically close species are more likely to compete with each other due to their shared traits and resource overlap, leading to less probability of their co-existence.

Due to the non-zero correlation between metabolic interactions and phylogenetic distances, comparing complementarity and competition between species pairs without accounting for their phylogenetic relationships confounds such comparisons. As an example, in Fig 2A, if a pair of closely related genomes (with phylogenetic distance close to 0) had complementarity index of 0.18 and therefore would be a significant outlier comparing to other pairs of genomes of similar phylogenetic distance. However, if this complementarity index was to be compared to those of genome pairs with greater phylogenetic distance (e.g., complementarity index of 2), it would no longer be considered a statistical outlier. To compare complementarity/competition indices across species, the confounding effects of phylogeny must be first decoupled.

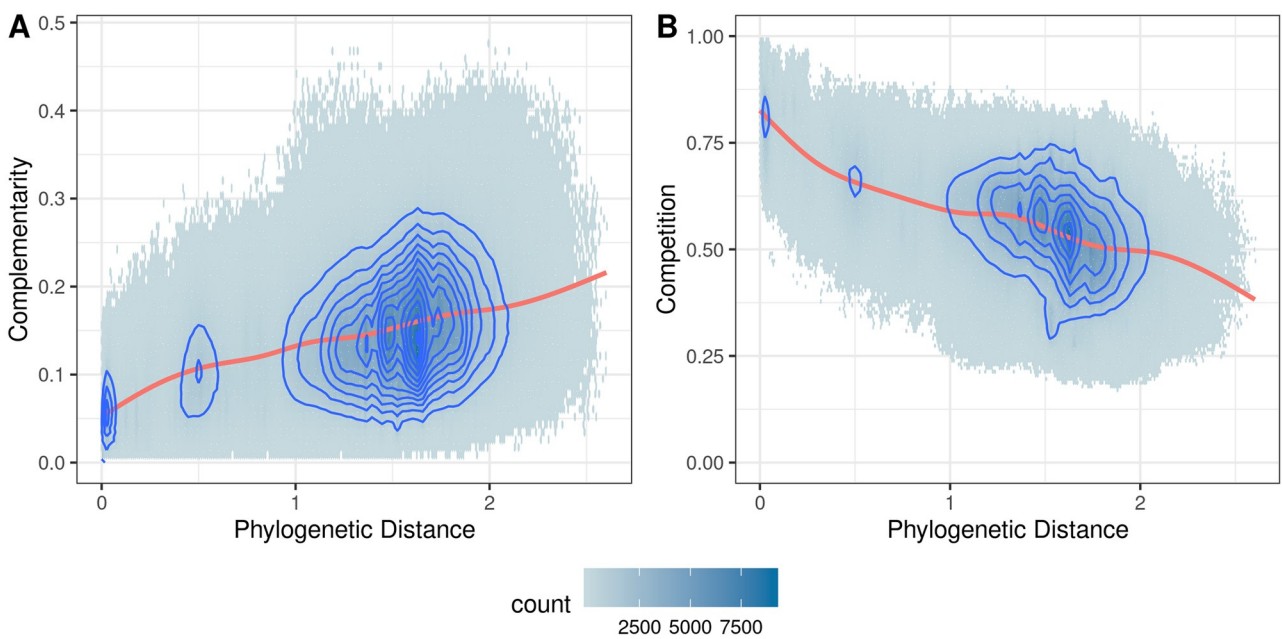

**Fig 2. Hexagonal binned plots of metabolic interaction indices versus phylogenetic distance.** Pairwise comparison between 2,815 human gut related MAGs (A) Metabolic Complementarity Index and (B) Metabolic Competition Index, versus their phylogenetic distance with density contours. The plots were fitted with a generalized additive model (red line).

Here we demonstrate a discretization approach for the identification of statistically significant complementary species pairs as a method for accounting/correcting for phylogenetic distances. To discretize comparisons across continuous phylogenetic distances, pairwise indices were binned by their phylogenetic distances. Outliers are then identified within each bin, which are likely pairs of bacteria with statistically significant complementary or competitive interactions.

## Identification of potentially collaborative or competing pairs of gut bacteria from metabolic outliers

To explore the relationship between complementary and competitive pairs, we compared their respective Z-scores (Fig 3). Significant outliers were selected using a Z-score threshold of ±2.698 as proposed by Tukey [39]. A total of 60,116 directed pairs were identified as positive complementary outliers. Additionally, 7,769 and 44,409 competitive positive and negative directed pairs of outliers were identified, respectively. Unsurprisingly, most pairs were centered around a Z-score of zero and no pairs were simultaneously significant for both complementarity and competition, simultaneously.

We analyzed bacteria pairs belonging to the same genus or family that have significantly high complementarity scores to better understand how taxonomic similarity correlates with metabolic cooperation. At the genus level, 140,152 directed pairs were identified; and at the family level, 233,555 directed pairs were identified. Of the pairs belonging to the same genus or family, 1,230 and 5,190 were identified as significant complementary outliers, respectively. These taxonomically similar bacteria pairs have the potential to cooperate in gut microbiomes. The rarity of significant outliers with the same taxonomic classifications suggests that for most taxonomically similar pairs at the genus and family level, niche differentiation plays an integral

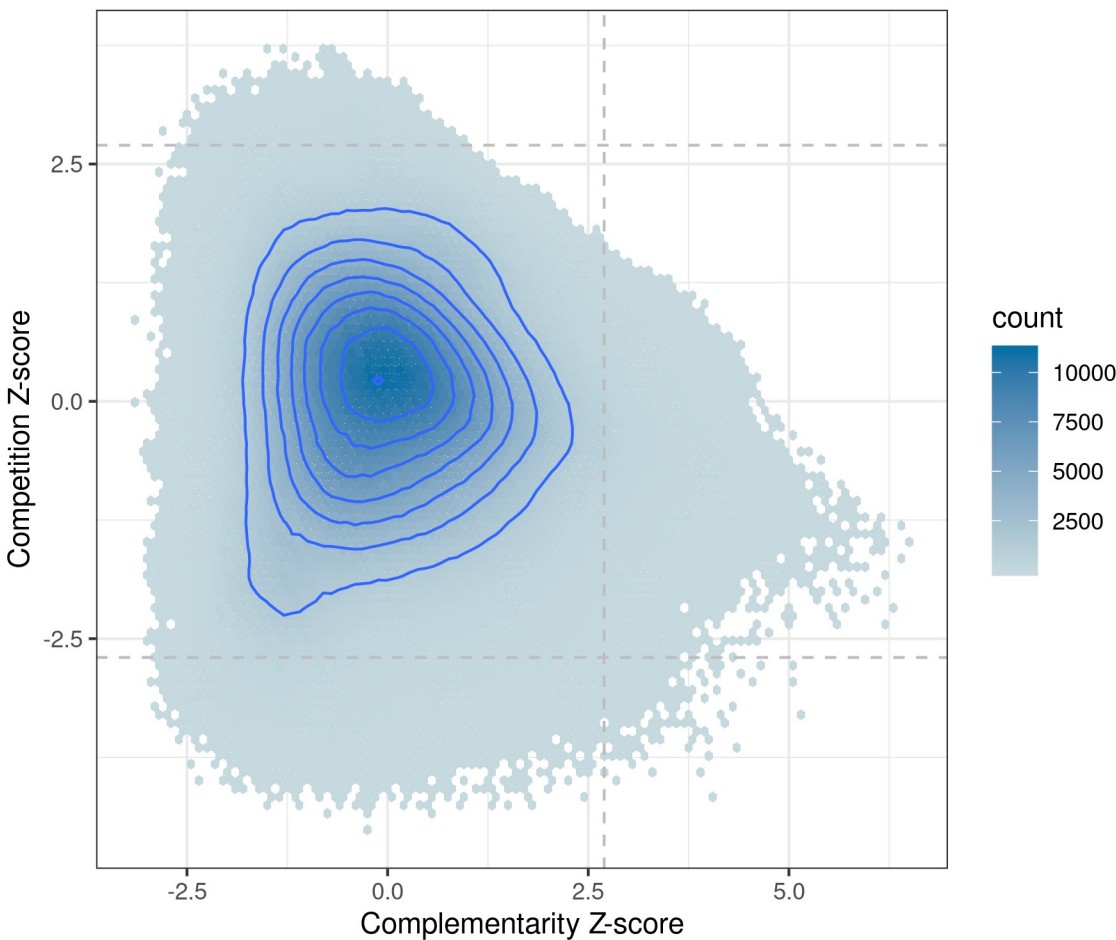

**Fig 3. Hexagonal binned plot of metabolic complementarity and competition Z-scores with density contours.**

role in community assembly. Detailed lists of complementarity and competition outliers are provided in S2 and S3 Tables, respectively.

To explore community assembly dynamics, we constructed a directed graph of bacterial species where bacteria are the nodes and a directed edge is added between two bacteria if they have a high metabolic complementarity (Z-score > 2.698) and low metabolic competition (Z-score < −1.000); here we relaxed the Z-score of competition indices to -1.000 in order to focus our analysis towards species pairs with greater complementarity while still constraining the analysis to a degree of low competition observed between species pairs (see S4 Table for detailed list). Using Infomap [40] to analyze the network, we were able to identify two main community modules (Fig 4). The larger community module (shown on the right in Fig 4) was populated with many multi-layer sub-modules, which featured majority of the significantly cooperating bacteria. Interestingly the smaller community module (shown on the left in Fig 4) exclusively contained *Bifidobacterium spp.* (e.g. *B. longum*, *B. bifidum*, *B. infantis*), suggesting that various *Bifidobacterium* species are metabolically complementary to each other, more-so than other phylogenetically similar taxa.

*Bifidobacterium* species are major colonizers of infant gut microbiota, and play a prominent role in the degradation and metabolism of Human milk oligosaccharides (HMOs) [41]. One such example of the complementary interactions between *Bifidobacterium* species was

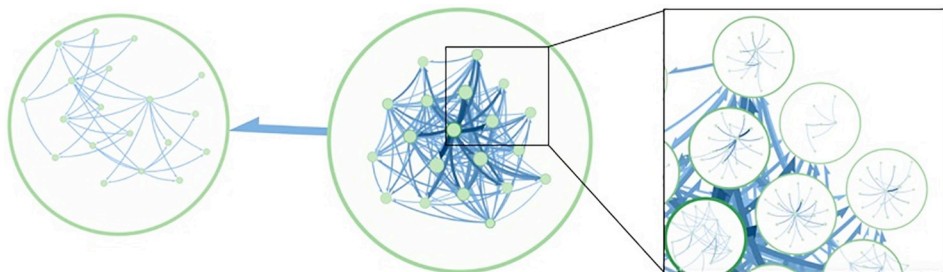

**Fig 4. Community modules of significant complementarity outliers that exhibit low metabolic competition identified from human-gut related MAGs.** Circular nodes represent predicted community modules and sub-modules of cooperative bacterial communities.

captured by our pipeline, and can be exemplified in predicted presence/absence of sialic acid metabolism pathways (Fig 5). The GEM for *B. infantis subsp. infantis* (Fig 5; left) was able to capture the pathways involved in sialic acid metabolism; whereas the GEM for *B. longum* (Fig 5; right) failed to capture any metabolic pathways that utilize sialic acid. While both species are present in high concentrations in the infant microbiota, various studies have shown that *B. longum* lacks the associated gene clusters for the sialic acid catabolism [42, 43]. It should be noted that while the GEM was able to capture sialic acid metabolic pathways in *B. infantis*, CarveMe failed to predict exo-$\alpha$-sialidase mediated degradation of sialylated carbohydrates. A

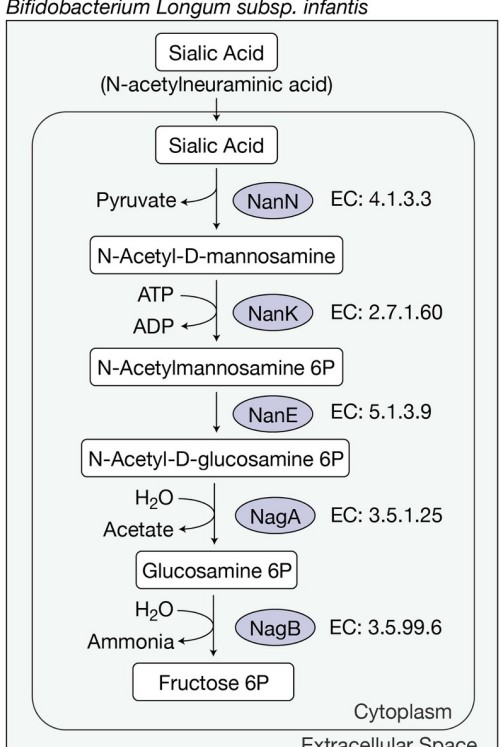

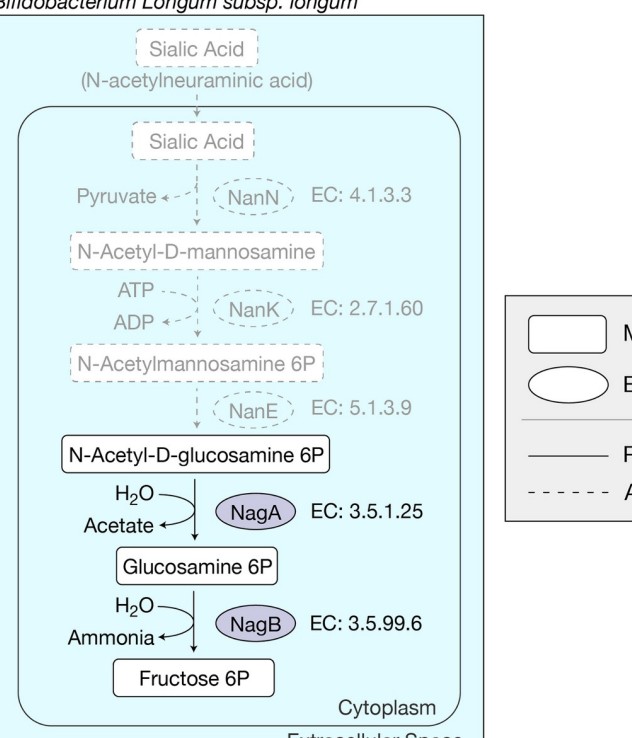

**Fig 5. Sialic acid metabolism pathway reconstructed from *Bifidobacterium species*.** (Left) Metabolic pathway of sialic acid metabolism present in the GENRE constructed from *B. infantis* MAG (GCF_000020425.1). (Right) Metabolic pathway of sialic acid metabolism present in the GENRE constructed from *B. longum* MAG (18391_1_6), missing metabolites and metabolic reactions from the GENRE model are grayed out with dotted lines.

protein blast of exo-$\alpha$-sialidase (NanH1 and NanH2) from *B. longum subsp. infantis* ATCC 15697 [44] against protein coding genes of *B. infantis* (MAG: GCF_000020425.1) was able to confirm the presence of both protein coding genes with 100% percent identity (S5 Table); both exo-$\alpha$-sialidase genes were absent in *B. longum* (MAG: 18391_1_6) (S6 Table). Nevertheless, this example shows a possible metabolic complementary between related species reflected within the GENREs. In addition to *Bifidobacterium spp*., other bacterial genera were shown to also form sub-community modules highly uniform for their own genera (i.e. *Helicobacter*, *Collinsella*, *Lachnospiraceae*, and *Ruminococcus*). We note that if complementarity scores were analyzed without correcting for phylogenetic distances, these significant complementarity scores of taxonomically similar bacteria would not be considered significant, thus emphasizing the importance of correcting for phylogenetic distances. The pattern of taxonomically related genomes forming community module is suggestive of habitat filtering characteristics within certain distinctive bacterial taxa. Infomap community module membership available in S1 File.

To further explore this, we analyzed the proportion of significantly cooperative bacteria with the same genus annotations. Our results show that more than half (42/76) of the taxa with 50 or more members within the same genus contained a significant number of metabolically complementary pairs; within genus proportion of taxa with significant pairs ranged from 0.02% to 15.9% (S7 Table). Together, these results show that while niche differentiation dominates a majority of metabolic interactions, we observe habitat filtering characteristics within certain bacterial taxa.

## Discussion

Here we demonstrate a novel approach to identifying significant metabolic cooperators and competitors between bacterial species pairs. This approach builds upon previously developed metrics of metabolic complementation and competition [34, 45, 46] by identifying outlier pairs relative to their phylogenetic distances. As pairwise metabolic interactions are correlated with phylogenetic distance, it remains imperative to take into consideration their phylogenetic distances when making comparisons across different phylogenetic distances as such comparisons may confound comparisons.

Our analysis shows that metabolic cooperation exhibits a positive relationship with phylogenetic distance, whereas metabolic competition exhibits a negative relationship. These findings support the results from previous work that studied the relationship between phylogenetic relatedness and gene content, functional distance, and metabolic interactions [24, 26, 38]. Together these observed relationships seem to support the theory of niche differentiation, where functional overlap discourages phylogenetically related species from co-existing. However, by taking into consideration the phylogenetic distance between pairs to identify metabolic outliers, we were able to identify significant intra-genus cooperation in several distinct taxa. The intra-genus modules may suggest that while most bacteria interactions display niche differentiation characteristics, some taxa exhibit habitat filtering. Notably, *Bifidobacterium* species were shown to form distinct community modules which suggest significant intra-genus cooperation compared to other taxa. These results support recent findings that suggest strains of *Bifidobacterium spp*. in infants have different nutrient profiles to support colonization of other specific *Bifidobacterium* species [47]. The observation of both habitat filtering and niche differentiation characteristics suggests that in some cases both contribute to the dynamics of community assembly.

We note a few limitations of our approach. First, metabolic complementarity and competition indices are dependent on a given metabolic model. Completeness of GENREs are

dependent on a variety of variables (e.g. the reconstruction tool and the genome completeness) that can have a significant impact on predicted metabolic interactions. Second, seed sets used to calculate the metabolic interaction indices do not represent required metabolites for growth, but rather represent a baseline of metabolites that in theory enable a given bacterium to produce any metabolite in their predicted metabolic network. As such, seed sets may influence the overestimation or underestimation of metabolic interactions between bacterial species. However, by integrating phylogenetic distances to normalize metabolic interaction indices, we believe that our approach provides a more accurate prediction of metabolic interactions in comparison to other similar methods. Additionally, low abundant microbial species within microbiomes are not always well represented within metagenomic samples but may play key roles within a metabolic network. While we acknowledge that validation of this method remains difficult due to the lack of a gold standard comparison, the non-independent nature of comparative metrics between organisms due to shared ancestry provides a logical explanation as to the necessity to account for such confounding effects.

The ability capture the presence/absence of sialic acid metabolism pathways within *Bifidobacterium spp*. MAGs provides an example of CarveMe's ability to reconstruct meaningful biological pathways. However, CarveMe's reliance on reference models fails to capture species and/or strain specific metabolic pathways absent from those utilized reference databases. This ultimately is a current limitation of automated GENREs, and a limitation of reference based techniques. Gap filling and manual curation of metabolic models can be also be used to supplement reconstruction of highly accurate models. With our method, metabolic model reconstructions can be easily interchanged and as new metabolic reconstruction tools are developed, the phylogenetic adjustment of Complementarity and Competition idicies can be easily applied when comparing metabolic networks.

By decoupling phylogenetic distances between Complementarity and Competition indicies, we provide a method to explore statistically significant cooperating/competing species pairs within microbbiomes to better understand community assembly dynamics. Additionally, competition networks can be used to identify highly competitive species pairs, which may be useful for suggesting beneficial probiotic candidates. A future research direction is to integrate phylogenetically-corrected complementarity and competition scores with co-occurrence information to better address the challenges of identifying bacterial interactions through mechanistic insight.

## Materials and methods

### Genome sequences of human-gut bacteria

To assemble the human-gut associated reference genomes, we collected genomes from two recent studies [48, 49]. Bacterial genomes reported in [49] were compiled from two sources: a total of 617 genomes obtained from the human microbiome project (HMP) [50], and 737 whole genome-sequenced bacterial isolates, representing the Human Gastrointestinal Bacteria Culture Collection (HBC). These 737 bacterial genomes were assembled by culturing and purifying bacterial isolates of 20 fecal samples originating from different individuals [49]. The bacterial genomes reported in [48] were generated and classified from a total of 92,143 metagenome assembled genomes (MAGs), among which a total of 1,952 binned genomes were characterized as non-overlapping with bacterial genomes reported. We were able to retrieve 612 out of 617 RefSeq sequences using the reported RefSeq IDs. We only included genomes with > 80% completeness and < 5% contamination (via CheckM [51]). Our final dataset for this study contains a total of 2,815 genomes/MAGs. Taxonomic annotation of these genomes/MAGs was done using GTDB-toolkit's least common ancestors approach [52].

## Genome scale metabolic network reconstructions and analysis

Genome-scale metabolic network reconstructions (GENREs) for all genomes were constructed using CarveMe [32] with default parameters. Coding sequences (CDSs) of all input genomes were generated using FragGeneScan [53] to be used as input for CarveMe. Briefly, CarveMe is a genome-scale metabolic model reconstruction tool which utilizes a universal model for a top-down approach to build GENREs. In contrast to conventional bottom-up methods which require well defined growth media, manual curation and gap-filling, the top down approach of CarveMe removes reactions and metabolites inferred to be not present in the manually curated universal template.

## Evaluation of incomplete metagenome assembled genomes

As we utilized MAGs with greater than equal to 80% CheckM [51] genome completeness, we simulated the ability of CarveMe to construct GENREs on incomplete genomes between 70-100% of the total CDSs at 5% intervals. To accomplish this, we utilized complete reference genomes obtained from NCBI RefSeq database. We then predicted protein coding sequences using Frag-GeneScan [53]. Using a custom script, we randomly selected (with repeats) sets of 3 neighboring CDSs until a specified interval of remaining CDSs remained. Neighboring CDSs were removed as a set, as genes are often missing together from assembled genomes due to uneven binning and sequencing, rather than a completely random process. We then used the remaining CDSs as input for CarveMe. This was repeated for 50 times at each 5% interval. The resulting GENRE were then used to construct a metabolic network with directed edges. The metabolic networks of simulated incomplete genomes were then compared with the corresponding metabolic networks constructed from complete genomes. To compare the differences between the constructed metabolic models, we assessed the Jaccard Index between the sets of edges and nodes in each metabolic model. Additionally, because the number of source and sink nodes are utilized for the computation of the Complementarity Index and Competition Index, we also assessed the effect of incomplete genomes on metabolic reconstruction using the number of source and sink nodes.

## Phylogenetic distance

To compute pairwise evolutionary distances between gut bacteria, we first inferred a phylogeny covering all participating genomes using FastTree [54]. It was shown that using more phylogenetic marker genes (e.g. a set of 16S ribosomal protein sequences from each organism) gives trees with higher-resolution than the 16S rRNA gene alone [55]. A total of 120 bacterial marker genes were used to infer these phylogeny. The 120 marker genes used are ubiquitous among bacterial species and are shown to occur as single copies and less susceptible to horizontal gene transfer [56]. Amino acid sequence of protein coding genes were searched using HMMER3 [57] against a 120 HMM model database of marker genes received from Pfam [58] and TIGRfam databases [59]. Similar to the approach in [56], sequences extracted from each HMM model were individually aligned using hmmalign [57], which were later concatenated to form the final alignment. Poorly aligned regions were removed from the concatenated alignment and a final phylogeny was inferred using FastTree under WAG + GAMMA models. From the inferred phylogenetic tree, the phylogenetic (evolutionary) distance between two nodes (i.e., species) can be calculated as the sum of all the branch lengths between them.

## Species interaction indexes

To estimate potential metabolic cooperation and competition between bacterial species, we need to know their nutritional profiles, which however are unavailable for most of the gut

bacteria. Similar to the approach reported in [34, 60], we use the compound seed set of each species as a proxy for its nutritional profile: the seed set of a metabolic network is defined as the minimal subset of the compounds that cannot be synthesized from other compounds in the network (due to lack of the corresponding enzymes, and hence are exogenously acquired) but their existence permits the production of all other compounds in the network.

We implemented a pipeline for computing metabolic interaction indices from genome sequences. Our pipeline uses a) CarveMe for building genome-scale metabolic models from genome sequences, b) NetworkX [61] to identity seed compounds, and c) our own implementation (in Python) of the approaches for computing metabolic competition and complimentary indices given two genome-scale metabolic models. We call our pipeline PhyloMInt (Phylogenetically-adjusted Metabolic Interaction indices).

**Seed set identification.** Utilizing NetworkX v2.2 [61], strongly connected components (SCC) within the GENREs are identified. Confidence levels are assigned for all compounds relative to their SCC size, where the confidence level (C) is denoted as:

$$C = \frac{1}{(Component\ Size)} \tag{1}$$

The confidence level is representative of the confidence that a given compound belongs to the seed set. A threshold of $C \geq 0.2$ was used to select compounds to be regarded as compounds part of a given 'seed set' of a given organism as specified by [60].

**Metabolic competition and complementarity indices.** Given two genome-scale metabolic models (GEMs) A and B, their Metabolic Competition Index ($MI_{Competition}$) is calculated as the fraction of A's seed set that is also in B's seed set, normalized by the weighted sum of the confidence score [34, 46]. $MI_{Competition}$ estimates the baseline metabolic overlap between two given metabolic networks.

$$MI_{Competition} = \frac{\sum C(SeedSet_A \cap SeedSet_B)}{\sum C(SeedSet_A)} \tag{2}$$

Metabolic Complementarity Index ($MI_{Complementarity}$) is calculated as the fraction of A's seed set that is found within B's metabolic network but not part of B's seed set, normalized by the number of A's seed set in B's entire metabolic network [34, 45]. $MI_{Complementarity}$ represents the potential for A's to utilize the potential metabolic output of B.

$$MI_{Complementarity} = \frac{|SeedSet_A \cap \neg SeedSet_B|}{|SeedSet_A \cap (SeedSet_B \cup \neg SeedSet_B)|} \tag{3}$$

A toy example for the calculation of Metabolic Competition and Complementarity indices has been provided in Fig 6. We note that the competition and complementarity indices are asymmetric.

## Phylogenetic normalization and outlier detection

Pairwise metabolic complementarity and competition indices between species pairs are plotted against their predicted phylogenetic distance. While methods of outlier detection for continuous data exists, local peaks and troughs of indices relative to phylogenetic distance make it difficult to identify local outliers. Thus, we utilize a binning approach to limit outlier detection to localized values. Both metabolic complementarity and competition indexes use a two-step binning process to bin pairwise observations, first by using a fixed phylogenetic distance interval of 0.01, followed by merging bins which are smaller than a prespecified size. Here we used the first bin size as the reference. Bins were merged with the closest preceding bin satisfying our

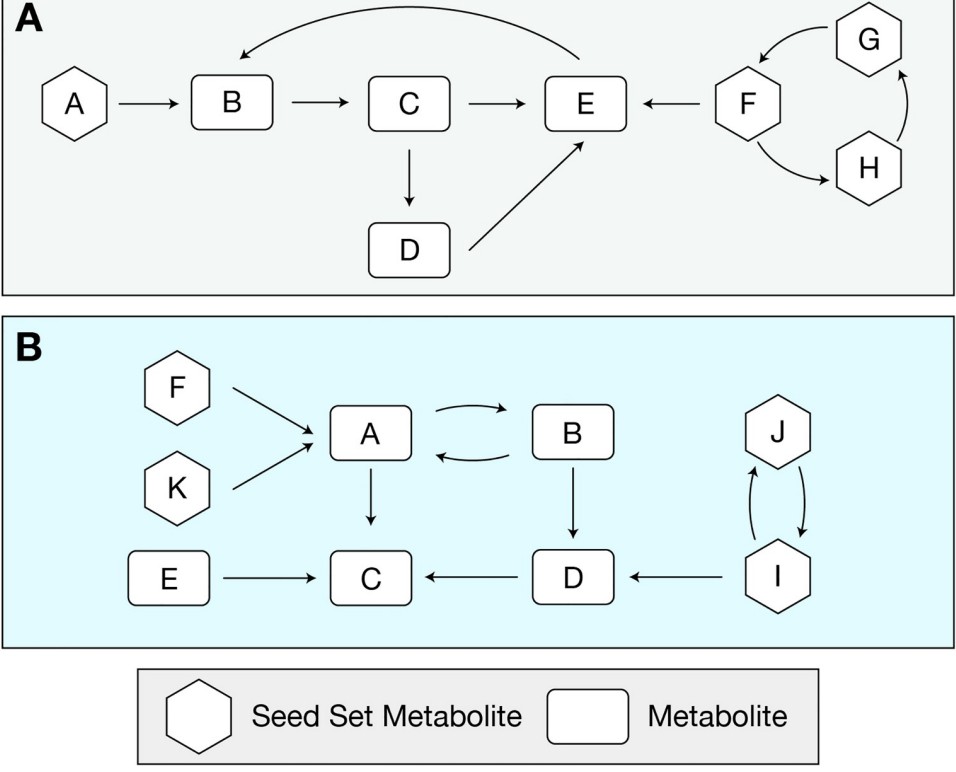

**Fig 6. Schematic illustration of seed set identification, and complementarity and competition index calculation between two toy metabolic networks.** In metabolic pathway A, $SeedSet_A$ consists of metabolites A, F, G, and H; metabolites F, G, and H form a strongly connected component (SCC). Confidence level of seed set metabolites within metabolic network A is $1, \frac{1}{3}, \frac{1}{3}$, and $\frac{1}{3}$ for metabolites A, F, G, and H, respectively. In metabolic pathway B, $SeedSet_B$ consist of F, I, J, and K; metabolites I and J form a SCC. Confidence level of seed set metabolites within metabolic network B is $1, \frac{1}{3}, \frac{1}{3}$, and 1 for metabolites F, I, J, and K, respectively. In a comparison between metabolic network A versus metabolic network B, metabolic network A shares only one seed metabolite with metabolic network B (metabolite F) which lies in the SCC in metabolic network A. Thus, the $MI_{Competition}$ between metabolic network A and B is $\left(\frac{1}{3} \div 2\right) = \frac{1}{6}$. Among $SeedSet_A$, metabolites A and F are found within the metabolic network B but only metabolite A is within non-$SeedSet_B$, thus the $MI_{Complementarity}$ index between metabolic network A and metabolic network B is 0.5. We used the same toy networks as those in [45].

minimum bin size threshold. To identify metabolic complementarity and competition outliers within each phylogenetic distance bin, we calculate the Z-score within each bin respectively. Tukey's method for outlier detection (equivalent to a Z-score threshold ±2.698) [39] was utilized to identify significant outliers.

## Network construction and community detection

To build a metabolic complementarity/competition network, species pairs are represented as nodes within the network. Identified significant outliers were used to construct a network of gut bacteria, in which for any pair of species A and B, a directed edge is added between A and B (from A to B), if A and B have significantly high complementarity score but low competition score. Using the adjacency list of the directed graph, a local installation of Infomap [40] (with the parameters: –directed –zero-based-numbering –num-trials 10) was utilized to identify community interaction modules within our dataset. Infomap is a random walk based approach

for community detection, and it provides a user friendly interface for visualization and exploration of the network and community structure (https://www.mapequation.org/navigator).

## Supporting information

**S1 File. Infomap output of community modules identified from metabolic complementarity index outlier network.**
(TXT)

**S1 Table. Metabolic network statistics of simulated incomplete genomes.**
(XLSX)

**S2 Table. Metabolic complementarity index outliers.**
(TSV)

**S3 Table. Metabolic competition index outliers.**
(TSV)

**S4 Table. Pairwise list of idnetified high metabolic complementarity (Z-score > 2.698) and low metabolic competition (Z-score < −1.000) bacteria.**
(TSV)

**S5 Table. Protein BLAST results.** Protein BLAST of exo-$\alpha$-sialidase (NanH1 and NanH2) from *B. longum subsp. infantis* ATCC 15697 [44] against protein coding genes of *B. infantis* (MAG: GCF_000020425.1).
(TSV)

**S6 Table. Protein BLAST results.** Protein BLAST of exo-$\alpha$-sialidase (NanH1 and NanH2) from *B. longum subsp. infantis* ATCC 15697 [44] against protein coding genes of *B. longum* (MAG: 18391_1_6).
(TSV)

**S7 Table. Analysis of the proportion of significantly cooperative bacteria within the same Genus.**
(TSV)

## Author Contributions

**Conceptualization:** Tony J. Lam, Yuzhen Ye.

**Data curation:** Tony J. Lam, Moses Stamboulian, Wontack Han.

**Formal analysis:** Tony J. Lam, Yuzhen Ye.

**Funding acquisition:** Yuzhen Ye.

**Methodology:** Tony J. Lam, Moses Stamboulian, Wontack Han.

**Project administration:** Tony J. Lam, Yuzhen Ye.

**Resources:** Tony J. Lam, Moses Stamboulian, Wontack Han.

**Software:** Tony J. Lam.

**Supervision:** Yuzhen Ye.

**Validation:** Tony J. Lam, Moses Stamboulian, Wontack Han.

**Visualization:** Tony J. Lam.

**Writing – original draft:** Tony J. Lam, Moses Stamboulian, Yuzhen Ye.

**Writing – review & editing:** Tony J. Lam, Moses Stamboulian, Yuzhen Ye.

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
