## [Decision Letter · Decision Letter 0]

10 Jun 2020

Dear Dr. Ye,

Thank you very much for submitting your manuscript "Pathway-based and phylogenetically adjusted quantification of metabolic interaction between microbial species" for consideration at PLOS Computational Biology.

As with all papers reviewed by the journal, your manuscript was reviewed by members of the editorial board and by several independent reviewers. In light of the reviews (below this email), we would like to invite the resubmission of a significantly-revised version that takes into account the reviewers' comments.

We cannot make any decision about publication until we have seen the revised manuscript and your response to the reviewers' comments. Your revised manuscript is also likely to be sent to reviewers for further evaluation.

Sincerely,

Christos A. Ouzounis

Associate Editor

PLOS Computational Biology

Jason Papin

Editor-in-Chief

PLOS Computational Biology

Reviewer's Responses to Questions

**Comments to the Authors:**

Reviewer #1: This paper is applying an original and advanced computational approach to address an extremely important, fundamentally and practically, problem of metabolic interactions, competitive and cooperative, in complex microbial communities. The developed software (computational pipeline PhyloMInt) combines existing tools for automated genome-scale metabolic reconstruction/modeling (CarveMe) and dissection of so called “seed” compounds (NetworkX) with the original implementation of an algorithm for computing of metabolic competition/cooperation indices. This comprises one of the most impactful aspects of the submitted MS. The availability of the developed software and all of the needed components at github is commendable, and the possibility for any reader to seamlessly install and even modify (if needed) the software add strongly to the anticipated impact of this publication! While the utility of the approach is demonstrated specifically for the human gut microbiome (HGM) (> 2,800 genomes including MAGs), it can be seamlessly expanded to any microbial community limited strictly by the availability of genomes and a quality of their functional annotations.

Additional strength of the paper is the demonstration of non-independence between computed metabolic indices and phylogeny, which is consistent with previously recognized trends of phylogenetically (and, thus, metabolically) close species to compete for the niche resources, and the opposite trend for phylogenetically distant organisms. This finding prompted authors to develop and implement an approach to minimization of this phylogenetic bias. Application of the developed pipeline to the analysis of HGM datasets leads to a reasonable (albeit not unexpected) conclusion that both types of metabolic interactions (and both drivers, “habitat filtration” and “niche differentiation”) play a role in HGM ecology.

Based on these considerations, this article is expected to attract a broad readership and have a notable impact in the field. However, prior to its publication, some work is required to mitigate several noted weaknesses briefly outlined below:

1. Ttile. The current title is somewhat misleading. Unlike MinPath, a previous very impactful accomplishment of the corresponding author, the PhyloMint pipeline, as well as the article per se, does not operate with a concept of metabolic pathways (a biologically solid and practically useful paradigm of predominant and semi-isolated segments of metabolic networks). Instead, it relies on a far more ambitious but less granular/accurate (albeit computationally more attractive) concept of automated genome-scale metabolic models. It is desirable to duly reflect it in the title to avoid confusion.

2. Supplementary data. They are mentioned several times in the MS (line 121, 162, 188, 194) and they are seemingly important but not properly organized and referred to. Instead of a generic umbrella reference in its current form (“details can be found in supplementary data”), which is not helpful at all, every reference should point to a specific supplementary table, graph or file.

3. A more important weakness of the paper is a lack of even one meaningful biological illustration of the main observations and conclusions. A very weak attempt to do so (p. 6, lines 177-183) is really not helpful. An interesting observation about possible metabolic complementarity of distinct species of Bifidobacteria actually has merit, which should be discussed explicitly (and not via a reference to a paper, which is also not very helpful in that sense). Examples of metabolic complementarity could have included: (i) B2 auxotrophy, which is characteristic for most Bifidobacteria except B. infantis, one of the earliest colonizers of breast milk-fed infants; (ii) ability/inability to utilize HMOs also distinguishing otherwise closely related B. longum and B. infantis; (iii) extracellular degradation of fucosylated and sialilated HMOs, which is uniquely characteristic of B. bifidum and may comprise a mechanism of metabolic cooperation with co-colonizing B. infantis. Considering this (or another) example implicated by PhyloMInt at some level of biological/biochemical details would strongly impact the interest of biologists to the methodology and conclusions. In fact, it would be important to openly state whether indeed at least some (or even one) of the known distinctive metabolic features were indeed picked up by CarveMe tool. [Off note, the fact that the quality of the PhyloMInt analysis may be only as good as underlying metabolic reconstructions that are full of errors, deficiencies and inconsistencies (which authors should have admitted while providing rationale for their choice of CarveMe over other overpromising automated model generators), does not diminish the importance of the developed downstream approaches and tools.]

4. In the same context: line 181- “…dominated by taxonomically similar genomes (Helicobacter, Collinsella, Lachnospiraceae, and Ruminococcus).” What was meant by this sentence? That Helicobacter is a close relative of Ruminococcus? Seriously? And if not, then what? And this is only one of the two sentences containing a reference to actual biology… The reader may lose all interest and confidence over vague statements like this one.

5. Providing one or two relevant metabolic examples would also allow authors to illustrate for the readers the entire conceptual apparatus of source and sink metabolites, seed compound sets and so forth that are very relevant for understanding of their analysis of metabolic interactions in mcirobial communities.

There is no doubt that authors will be able to quickly address these minor concerns (may be with a help of an expert in microbial metabolism), which would enhance the well-anticipated impact of this publication.

Reviewer #2: Predicting species interactions among a microbial community from genomic sequences is a challenging problem. Previous work has established algorithms to construct metabolic networks from genomics sequences and compute competition/complementary indices. This work further explored the impact of incomplete genomes assembled from metagenome and the presence of phylogenetically related clusters. Based on simulations the author concludes that genome completeness is unlikely to be factor to infer species interaction. Using a human microbiome dataset the authors normalized phylogenetic relatedness via a bin-based approach. Outliers from each bin are regarded as true interactions. The manuscript is overall easy to follow despite of some grammar errors.

I really like the directions the authors were taking, but feel the presented research is preliminary. More experiments are required to boost the rigor of this presentation. I also understand so far there are no control datasets with known species interactions for a robust estimation of the prediction accuracy, so one might have to be creative to come up with some metrics.

Some ideas to strengthen the paper:

1. While the source and sink nodes are necessary for metabolic network construction, the authors need to show that having incomplete genomes does not impact the resulting networks, not just the number of nodes. Moreover, Metagenome assembled genomes may not miss genes randomly, genes co-located in the same genomic region tend to be missed together (a binning problem).

2. The advantage of controlling phylogenetic relatedness was not clearly shown. Are there other independent evidence that may corroborate the statement that normalized indices are better than those without normalization?

3. In the method section, "Metabolic competition and complementarity indices", is the method the same as that presented in Levy and Borenstein (2013, ref 30)? If not, please list and justify the innovations here.

4. In the method section, "Phylogenetic distance", it seems that the authors presented a novel method. Has it been compared to established method, such as gANI (PNAS 2005;102:2567–2572) or GTDB-tk (https://doi.org/10.1093/bioinformatics/btz848)?

**Have all data underlying the figures and results presented in the manuscript been provided?**

Reviewer #1: No: Supplementary data are not properly organized. See comments about additional illustrations above.

Reviewer #2: Yes

PLOS authors have the option to publish the peer review history of their article (what does this mean?). If published, this will include your full peer review and any attached files.

Reviewer #1: No

Reviewer #2: No
---

## [Decision Letter · Decision Letter 1]

10 Sep 2020

Dear Dr. Ye,

We are pleased to inform you that your manuscript 'Model-based and phylogenetically adjusted quantification of metabolic interaction between microbial species' has been provisionally accepted for publication in PLOS Computational Biology.

Best regards,

Christos A. Ouzounis

Associate Editor

PLOS Computational Biology

Jason Papin

Editor-in-Chief

PLOS Computational Biology

Reviewer's Responses to Questions

**Comments to the Authors:**

Reviewer #1: I am very pleased with responses and revisions.

The paper was much improved and can be accepted as is.

Noted only one seeming typo:

Line 217, page 7:

"...example shows a possible metabolic complementary between related species reflected".

I assume it should be either "metabolic complementarity" or "complementary metabolic interactions"

Reviewer #2: The authors sufficiently addressed my previously concerns. I was glad to read the much improved manuscript.

**Have all data underlying the figures and results presented in the manuscript been provided?**

Reviewer #1: Yes

Reviewer #2: Yes

PLOS authors have the option to publish the peer review history of their article (what does this mean?). If published, this will include your full peer review and any attached files.

Reviewer #1: No

Reviewer #2: **Yes: **Zhong Wang

---

## [Editor Report · Acceptance letter]

21 Oct 2020

PCOMPBIOL-D-20-00796R1 

Model-based and phylogenetically adjusted quantification of metabolic interaction between microbial species

Dear Dr Ye,

I am pleased to inform you that your manuscript has been formally accepted for publication in PLOS Computational Biology. Your manuscript is now with our production department and you will be notified of the publication date in due course.

With kind regards,

Matt Lyles
